# A new 'CFS tracking' paradigm reveals uniform suppression depth regardless of target complexity or salience

**David Alais[1†], Jacob Coorey[1†], Randolph Blake[2], Matthew J Davidson[1]***

[1]School of Psychology, The University of Sydney, Sydney, Australia; [2]Department of Psychology, Vanderbilt University, Nashville, United States

**Abstract** When the eyes view separate and incompatible images, the brain suppresses one image and promotes the other into visual awareness. Periods of interocular suppression can be prolonged during continuous flash suppression (CFS) – when one eye views a static 'target' while the other views a complex dynamic stimulus. Measuring the time needed for a suppressed image to break CFS (bCFS) has been widely used to investigate unconscious processing, and the results have generated controversy regarding the scope of visual processing without awareness. Here, we address this controversy with a new 'CFS tracking' paradigm (tCFS) in which the suppressed monocular target steadily increases in contrast until breaking into awareness (as in bCFS) after which it decreases until it again disappears (reCFS), with this cycle continuing for many reversals. Unlike bCFS, tCFS provides a measure of suppression depth by quantifying the difference between breakthrough and suppression thresholds. tCFS confirms that (i) breakthrough thresholds indeed differ across target types (e.g. faces vs gratings, as bCFS has shown) – but (ii) suppression depth does not vary across target types. Once the breakthrough contrast is reached for a given stimulus, all stimuli require a strikingly uniform reduction in contrast to reach the corresponding suppression threshold. This uniform suppression depth points to a single mechanism of CFS suppression, one that likely occurs early in visual processing because suppression depth was not modulated by target salience or complexity. More fundamentally, it shows that variations in bCFS thresholds alone are insufficient for inferring whether the barrier to achieving awareness exerted by interocular suppression is weaker for some categories of visual stimuli compared to others.

**\*For correspondence:**
mjd070@gmail.com

[†]These authors contributed equally to this work

**Competing interest:** The authors declare that no competing interests exist.

## eLife assessment

This **valuable** study introduces an innovative method for measuring interocular suppression depth, which implicates mechanisms underlying subconscious visual processing. The evidence is **solid** in suggesting that the new method yields provocative uniform suppression depth results across image categories that differ from conventional bCFS threshold. It will be of interest not only to cognitive psychologists and neuroscientists who study sensation and perception but also to philosophers who work on theories of consciousness.

## Introduction

The quest to understand visual processing outside of awareness is tantalising but notoriously challenging to implement (*Breitmeyer, 2015*; *Hesselmann and Moors, 2015*; *Holender, 1986*; *Logothetis, 1998*; *Newell and Shanks, 2014*; *Schmidt, 2015*). There are controversial claims of higher level semantic processing (*Lanfranco et al., 2023*; *Mudrik et al., 2014*; *Stein and Sterzer, 2014*; *Sterzer et al., 2014*), object categorization (*Kouider and Dehaene, 2007*; *Rees, 2007*; *Sterzer*

*et al., 2014*), and abstract reasoning (*Hassin, 2013*; *Sklar et al., 2012*) occurring outside of awareness. A number of methods exist for manipulating visual awareness (*Kim and Blake, 2005*), with two of the most popular in recent years being binocular rivalry (*Alais and Blake, 2014*) and continuous flash suppression (CFS: *Fang and He, 2005*; *Tsuchiya and Koch, 2005*). Both rely on interocular suppression induced by dissimilar monocular images independently presented to each eye, the result being that only one eye's image is perceived (i.e. dominates) at any given moment, with the other suppressed from awareness. For images approximately matched in salience, as often is the case in binocular rivalry (*Alais and Blake, 2014*; *Wang et al., 2023*), monocular suppression lasts for just a few seconds before switching to suppress the other eye (and so on, alternating irregularly over time). With CFS, a highly salient stream of dynamic images seen by one eye suppresses a smaller, weaker target presented to the other for considerably longer periods of suppression that can last tens of seconds. The potency of CFS has great appeal when it comes to assessing residual effectiveness of different categories of visual stimuli blocked from awareness by CFS.

A commonly used variant known as 'breaking CFS' (bCFS) was introduced by *Jiang et al., 2007* in which the suppressed target slowly ramps up from low contrast until it becomes sufficiently strong to break suppression and achieve visibility. Time to breakthrough has become a popular measure, and differences in breakthrough times among various image types have been used to support claims for unconscious processing of certain visual images. According to this line of reasoning, image types that consistently emerge from suppression more rapidly than do other image types are construed to be ones that receive prioritized processing outside of awareness. To give three examples of this purported effect, faces portraying emotional expressions break suppression faster than do neutral faces (*Alais, 2012*; *Jiang et al., 2007*; *Tsuchiya et al., 2009*; *Yang et al., 2007*), complex scenes containing incongruent objects break suppression faster than do congruent scenes (*Mudrik et al., 2011*), and native words break suppression faster than do unfamiliar foreign words (*Jiang et al., 2007*). Sceptics argue, however, that differences in breakthrough times can be attributed to low-level factors such as spatial frequency, orientation and contrast that vary between images (*Gayet et al., 2014*; *Moors and Hesselmann, 2018*; *Moors et al., 2016*; *Moors et al., 2017*; Moors & Hesselmann, 2018; *Stuit et al., 2023*), and more fundamentally, that breakthrough times alone are insufficient to measure differential unconscious processing (*Stein and Hesselmann, 2019*).

Conclusions based on a comparison of bCFS breakthrough times between different image categories suffer from a problem of unidirectionality and a false assumption of image equivalence (an exception here is when the same image is compared under different contexts, such as after associative learning of value [*Lunghi and Pooresmaeili, 2023*] or fear conditioning [*Gayet et al., 2016*]). Images producing faster breakthrough times (equivalently, lower breakthrough contrasts) are interpreted as undergoing residual processing outside of awareness, adding to their salience and weakening their interocular suppression (*Mudrik et al., 2011*; *Sterzer et al., 2011*; *Zhou et al., 2010b*; *Zhou et al., 2010a*). An implicit assumption here is that as all images were initially invisible, the depth of interocular suppression was thus weaker for images that break suppression more quickly (i.e. produce faster reaction times, as sometimes stated). Yet, the depth of interocular suppression is rarely measured in CFS paradigms (see *Tsuchiya et al., 2006* for an exception), and to our knowledge, has not been explicitly compared between image categories.

One method for measuring interocular suppression is to compare the threshold for change-detection in a target when it is monocularly suppressed and when it is dominant, an established strategy in binocular rivalry research (*Alais, 2012*; *Alais et al., 2010*; *Alais and Melcher, 2007*; *Nguyen et al., 2003*). Probe studies using contrast as the dependent variable for thresholds measured during dominance and during suppression can advantageously standardise suppression depth in units of contrast within the same stimulus (e.g. *Alais and Melcher, 2007*; *Ling et al., 2010*). Ideally, the change should be a temporally smoothed contrast increment to the rival image being measured (*Alais, 2012*), a tactic that preludes abrupt onset transients and, moreover, provides a natural complement to the linear contrast ramps that are standard in bCFS research. In this study, we measure bCFS thresholds as the analogue of change-detection during suppression, and as their complement, record thresholds for returns to suppression (reCFS). We do this by including a bidirectional measure in which the target contrast steadily decreases (ramps down) over time, eventually culminating in the target's transition from dominance to suppression. By comparing the thresholds for a target to transition into (reCFS) and out of suppression (bCFS), we recognise that the criterion for change is much higher,

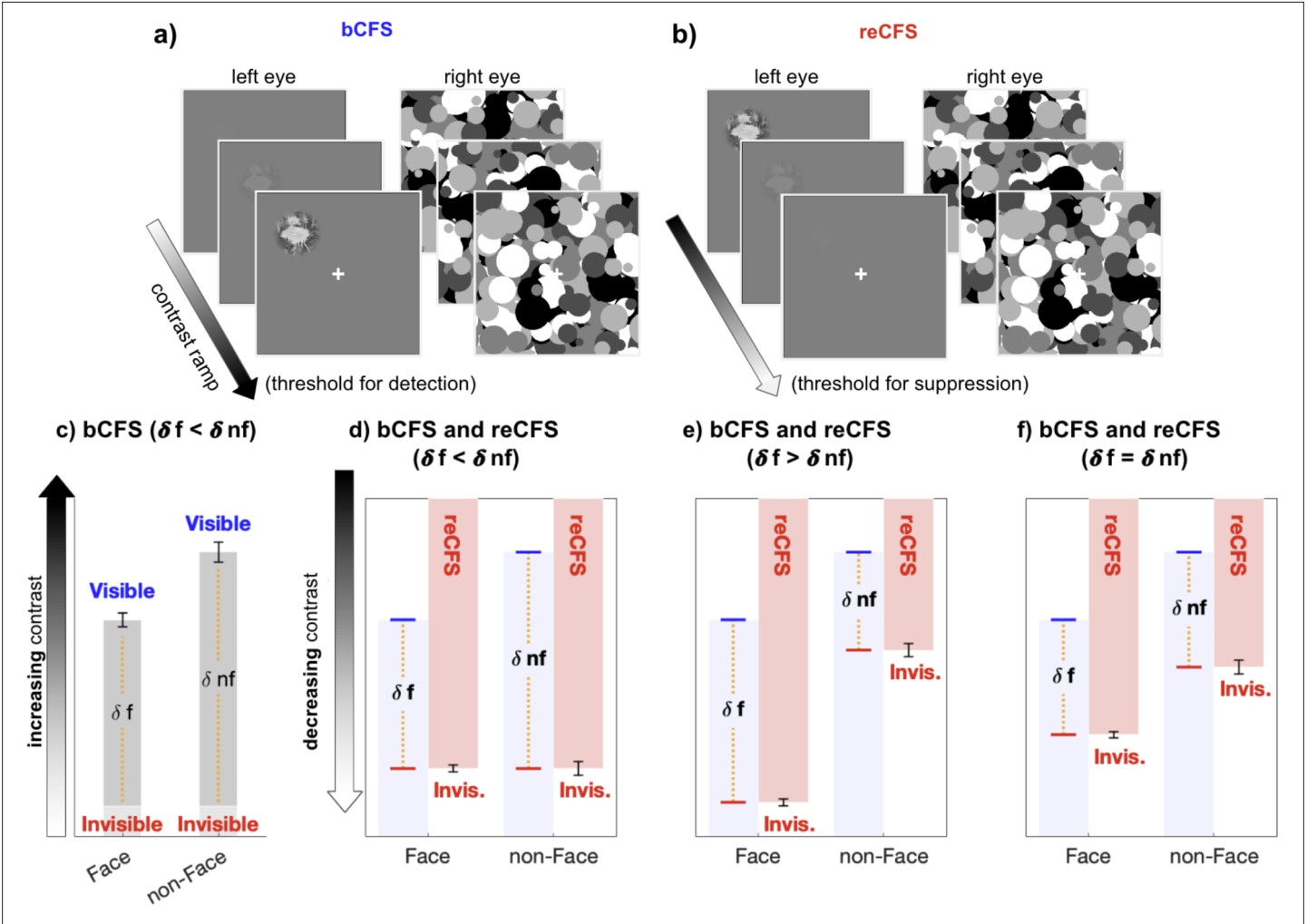

**Figure 1.** The critical measure of suppression depth remains untested in CFS. (**a, b**) Exampe stimulus displays for the discrete trials, during which (**a**) a low-contrast target steadily increases in contrast until visibility is reported (bCFS), and (**b**) discrete reCFS trials, with a high-contrast target decreasing until target invisibility is reported. (**c**) A typical bCFS result, in which a target image (here a face or non face) is initially weak and steadily increases until it breaks suppression. Faster breakthrough times (lower contrast) for face stimuli are often interpreted as evidence that faces undergo expedient processing that counteracts their susceptibility to suppression relative to other visual stimuli. Without also measuring suppression thresholds for each stimulus, this conclusion is premature. (**d-f**) To define the magnitude of suppression during CFS, it is necessary to measure the contrast thresholds at which stimuli enter and exit awareness, with the difference indicating suppression depth. Red bars display hypothetical results measuring the contrast at which an initially visible stimulus with decreasing contrast becomes suppressed by the mask (reCFS). The results of panel c are reproduced in d-f in light blue. (**d**) If the reCFS thresholds for face and non-face images are the same, this would support reduced suppression depth for faces (as δface < δ non-face). (**e**) Alternatively, the reCFS thresholds for faces and non-faces might differ, with the face remaining visible at a lower contrast than non-face images (lower reCFS threshold), indicating more suppression for faces than non-faces (as δface > δnon-face). (**f**) Finally, the reCFS thresholds might differ between faces and non-faces but by an amount equivalent to their bCFS differences, indicating the same suppression depth for both image types (as δface = δnon-face). Such a result would argue against enhanced unconscious processing of face stimuli.

and as a result expect larger suppression depths (on average) than those recorded in rivalry research. More importantly, however, by averaging the difference between bCFS and reCFS thresholds, a key analogue of suppression depth that has been missing from the CFS literature can be provided and compared between image categories. *Figure 1* outlines hypothetical results that can be obtained when recording reCFS thresholds as a complement to bCFS thresholds in order to measure suppression depth.

Here, we introduce a novel method termed 'tracking CFS' (tCFS) which combines alternating down-ramped and up-ramped targets, to provide a measure of suppression depth that allows a rigorous test of claims that certain image types undergo less suppression than others (see *Figure 2*).

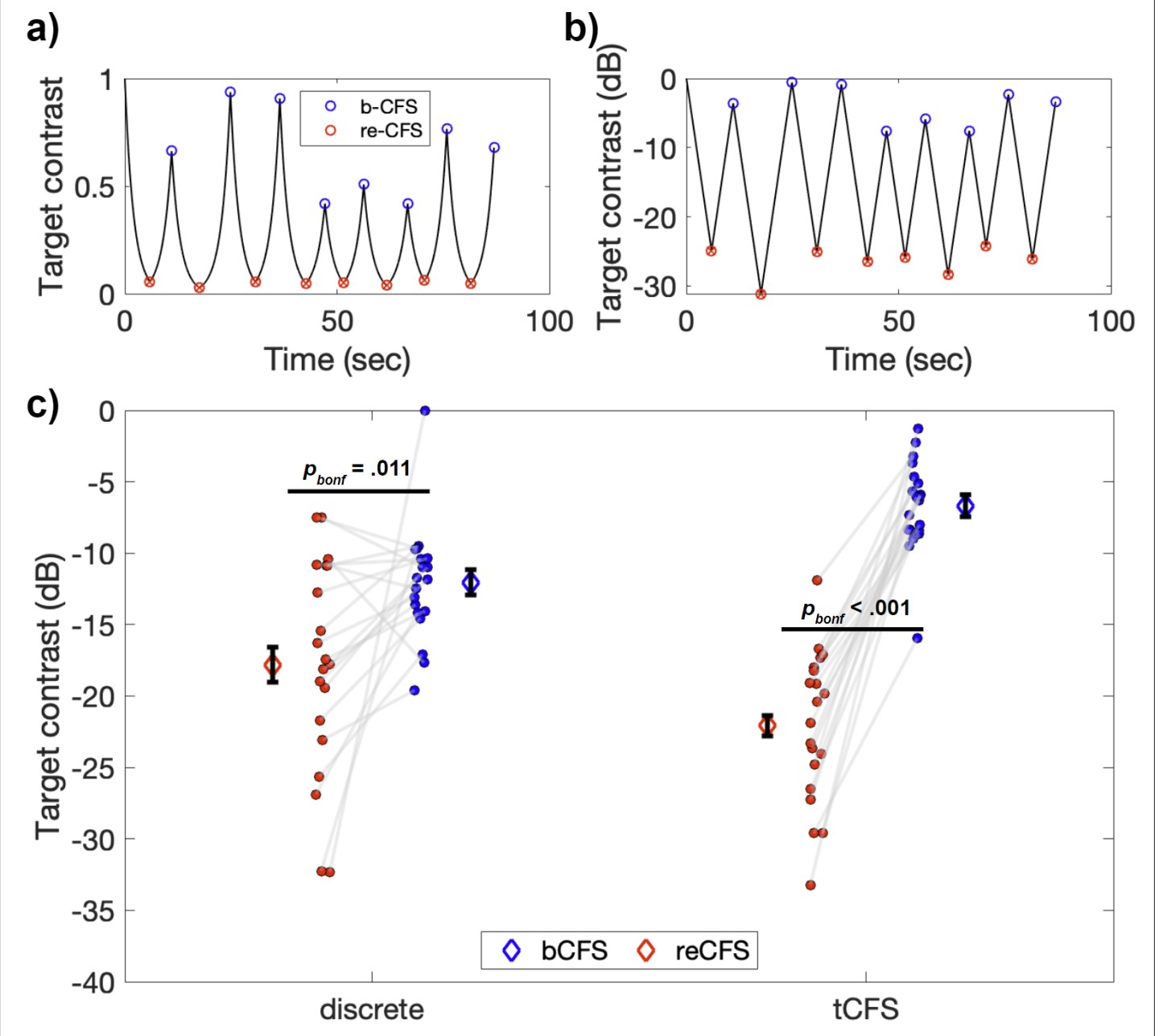

**Figure 2.** Example tCFS trial and comparison to discrete conditions. (**a–b**) Example tCFS trial from one participant, showing the change in contrast over time. Red markers indicate the level at which a target with decreasing contrast became suppressed (reCFS) and blue markers indicate traditional bCFS responses (breakthrough of contrast-increasing target). The same trial is shown with target contrast in decibel scale in b. Because the human visual system has a logarithmic contrast response, it is appropriate to increase/decrease target contrast logarithmically as in b, to create a contrast change that is perceptually linear. (**c**) Interaction between thresholds and condition type for the 20 participants. Individual blue and red dots display participant means for bCFS and reCFS respectively. Grey lines link thresholds per participant, per condition. Blue and red diamonds display the mean across participants, and error bars plot ± 1SEM corrected for within participant comparisons (**Cousineau, 2005**).

The participant views a visible target as it declines in contrast until it becomes suppressed. When suppression is reported the contrast change reverses direction and the participant indicates when the target emerges out of dominance and is seen once again (triggering a contrast decrease again, and so on over many reversals). By continually tracking thresholds over time for breakthroughs (bCFS) and succumbing (reCFS) to suppression, tCFS is very efficient, makes no assumptions of image equivalence, and provides inherently bidirectional measuresof CFS.

To preview the results, across three experiments we find no evidence using the tCFS paradigm for differences in suppression depth among several image categories (i.e. faces, objects, several types of gratings, and visual noise). We do replicate the common observation that breakthrough thresholds differ across image categories, but now we find that reCFS thresholds covaried with bCFS thresholds across all image categories, meaning that suppression depth was constant for faces, objects, gratings, and visual noise. In other words, we find no evidence to support differential unconscious processing among these particular, diverse categories of suppressed images.

## Results

### Experiment 1

Many previous bCFS studies have shown that *increasing* contrast eventually causes a target image to overcome interocular suppression (i.e. emerge into awareness). To our knowledge, however, none has investigated the contrast at which an initially visible image succumbs to suppression as its contrast is *decreased*. Experiment 1 uses the tCFS method to continuously track changes in target visibility as it rises and falls in contrast. As shown in *Figure 2b*, this provides a series of bCFS thresholds (upper turning points) as well as thresholds for the target's re-suppression (lower turning points: reCFS thresholds). We compare these thresholds to those obtained with a discrete procedure in which contrast either increased steadily from a low starting point until breakthrough (standard bCFS measure) or decreased steadily from a high starting point until suppression was achieved. On each trial, the target image was either a face or a familiar object, with the images all matched in size, RMS contrast and mean luminance.

A repeated-measures ANOVA revealed a significant main effect of threshold, such that bCFS thresholds were significantly higher than reCFS thresholds ($F(1,19) = 50.39$, p<0.001, $\eta_p^2 = 0.73$). There was also a significant interaction between threshold and condition ($F(1,19) = 41.19$, p<0.001, $\eta_p^2 = 0.68$), indicating that the difference between bCFS and reCFS thresholds was influenced by whether they were recorded using the discrete trial procedure or the continuous trial procedure. Subsequent post-hoc tests revealed that bCFS and reCFS thresholds differed within each type of procedure (Discrete: $t(19) = 2.89$, p=0.009, $d$=0.65 Continuous: $t(19) = 12.12$, p<0.001, $d$=2.7). Overall, the suppression depth (i.e. the difference between bCFS and reCFS thresholds) was larger with the continuous procedure ($M$=–15.40 dB, $SD$ = 5.68) compared to the discrete procedure ($M$=–5.74 dB, SD = 8.89), $t(19)$ = 6.42, p<0.001, $d$=1.3, a finding we return to in the Discussion. *Figure 2c* displays a summary of these results.

After confirming a difference between bCFS and reCFS thresholds, we next compared suppression depth measured using different image types. We repeated the analysis with the additional exploratory factor of image type (face vs object) in a 2x2 x 2 repeated-measures design (threshold, procedure, image type). We again found significant main effects of threshold ($F(1,19) = 50.39$, p<0.001, $\eta_p^2 = 0.73$), and a threshold x condition interaction ($F(1,19) = 41.19$, p<0.001, $\eta_p^2 = 0.68$), but non-significance for the effect of image type (p=0.26).

We quantified the evidence for this null-effect on suppression depth with a subsequent Bayesian model comparison. A Bayesian repeated-measures ANOVA (2x2; procedure x image type on suppression depth) found that the best model to explain suppression depth included the main effect of procedure ($BF_{10}$=3231.74), and weak evidence/data insensitivity for image type ($BF_{10}$=0.37). This indicates that the data was insensitive as to whether image-type was better at predicting suppression depth than the null model. In other words, there is a large disparity in contrast between targets breaking and re-entering CFS, but the magnitude of this disparity is most likely the same for objects and for faces. This potentially important finding is the focus of our next Experiment.

### Experiment 2

Experiment 1 disclosed that bCFS thresholds were not equivalent to reCFSthresholds, and demonstrated that the suppression depth of images could be quantified in an image-specific manner. Furthermore, Experiment 1 showed that suppression depth could be measured rapidly using the tCFS procedure and did not differ between faces and objects. In Experiment 2, we tested whether the constant suppression depth obtained in Experiment 1 for two image types would replicate across a larger variety of image categories.

We measured suppression depth for faces, objects, linear gratings, phase scrambled images, and radial/concentric patterns using the tCFS method. All have been used to investigate bCFS thresholds before (*Stein and Hesselmann, 2019*). We reasoned that faces and objects are familiar, meaningful stimuli processed widely throughout the visual hierarchy. Gratings and phase-scrambled images, on the other hand, comprise simpler, meaningless stimuli devoid of perceptual relevance. Phase-scrambled images were created from the object stimuli by randomising their phase spectra, rendering the images unrecognizable (e.g. *Piotrowski and Campbell, 1982*; *Yu and Blake, 1992*). These images thus served as controls for the regular stimuli, since both ordinary and phase-scrambled images have approximately the same Fourier energy spectra. Finally, polar patterns (radial and concentric gratings) were included because they have robust structural coherence that is registered by extrastriate visual areas in the ventral stream (*Wilkinson et al., 2000*), but are lacking in semantic meaning (*Hong, 2015*).

A 2x5 repeated-measures ANOVA revealed a significant main effect of threshold (bCFS vs reCFS; $F(1,17) = 133.79$, $p<0.001$, $\eta_p^2 = 0.89$), and image type ($F(4,68) = 13.45$, $p<0.001$, $\eta_p^2 = 0.44$). Critically however, there was no significant interaction between thresholds and image type ($p=0.98$), indicating that the relationship between bCFS and reCFS thresholds was invariant across image categories. A Bayesian repeated-measures ANOVA (1x5, effect of image categories on suppression depth), confirmed strong evidence in favor of the null hypothesis ($BF_{01} = 20.30$).

This result is plotted in *Figure 3*, which clearly shows differences in bCFS thresholds (blue symbols in *Figure 3a*) over image type (1x5 repeated-measures ANOVA; $F(4,68) = 16.29$, $p<0.001$, $\eta_p^2 = 0.49$), as has been reported in many studies. Scrambled noise images, for example, yield breakthrough thresholds 5.3 dB higher than do face images, and linear gratings breakthrough thresholds are 3.0 dB higher than polar gratings. Critically, *Figure 3* also shows that reCFS thresholds exhibit the same pattern of differences over image type (red symbols in *Figure 3a*): and as confirmed by our Bayesian model comparison, the depth of suppression is constant across all image categories.

Replicating the result of Experiment 1, approximately 15 dB of suppression depth was observed in Experiment 2 (*Figure 3b*), which was practically identical across image types (Faces, $M=14.35$ (SD = 5.64); Objects, $M=14.61$ (SD = 6.07); Gratings, $M = 14.88$ (SD = 5.55); phase-scrambled images, $M=14.70$ (SD = 5.39)); polar patterns, $M=14.69$ (SD = 6.50). A repeated-measures ANOVA confirmed that suppression depth did not differ across image categories ($F(4,68) = 0.1$, $p=0.98$).

## Experiment 3

Experiments 1 and 2 introduced the tCFS method and, using that new method, demonstrated that bCFS thresholds vary depending on target type, as many previous bCFS studies have shown. Importantly, reCFS thresholds varied in parallel with bCFS thresholds which, when expressed in terms of suppression depth, reveals that depth of suppression is strikingly uniform across target image categories in CFS. This uniformity of suppression depth could indicate that neural events mediating CFS suppression transpire within a common visual mechanism, one that is not selective for image type, complexity or semantic meaning. This view is compatible with popular models of binocular rivalry built around the concept of reciprocal inhibition and neural adaptation (*Alais et al., 2010*; *Kang and Blake, 2010*; *Mcdougall, 1901*), as well as with more recent Bayesian-inspired inference-based models in which perceptual alternations in dominance are triggered by accumulating residual error signal associated with competing stimulus interpretations (*Hohwy et al., 2008*). As pointed out elsewhere (*Blake, 2022*), steadily increasing error signal effectively plays the same role as does steadily decreasing inhibition strength caused by neural adaptation in reciprocal inhibition models.

In the context of the tCFS method, the steady increases and decreases in the target's *actual* strength (i.e. its contrast) should, respectively, boost its emergence from suppression (bCFS) and facilitate its reversion to suppression (reCFS) as it competes against the mask. Whether construed as a consequence of neural adaptation or error signal, we surmise that these cycling state transitions defining suppression depth should be sensitive to the rate of contrast change of the monocular target. Specifically, the slower the contrast change, the greater the amount of accrued adaptation, which will contract the range between breakthrough and suppression thresholds according to an adapting reciprocal inhibition model. For fast contrast change, there will be less accrual of adaptation meaning that the range between breakthrough and suppression thresholds will exhibit less contraction. Expressed in operational terms, the depth of suppression should be positively related to the rate of target change. Experiment 3 tested this supposition using three rates of contrast change.

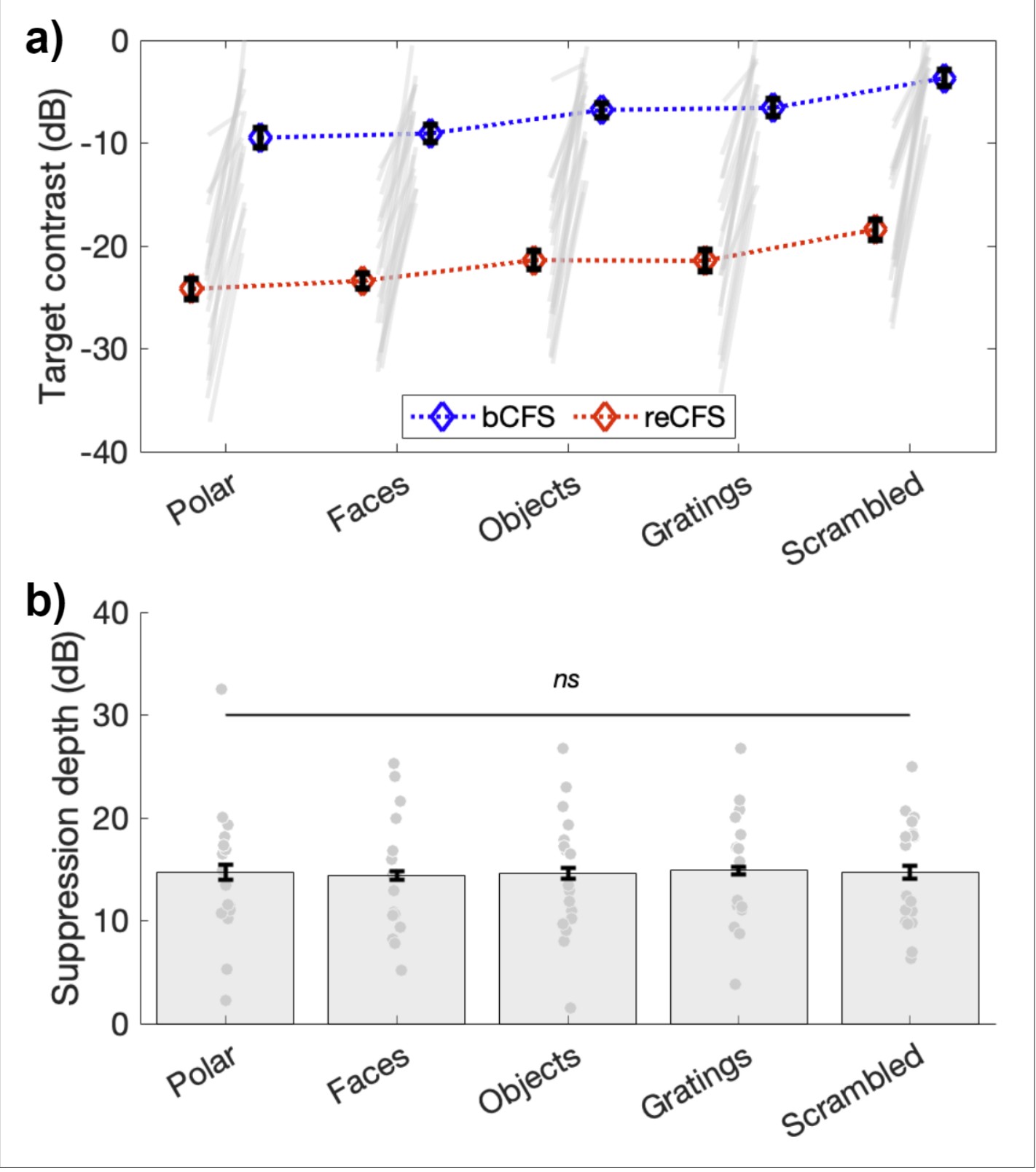

**Figure 3.** Suppression depth is uniform across image categories. (**a**) bCFS and reCFS thresholds by image type for the 18 participants. Blue and red diamonds display the mean across participants, and error bars plot ±1 SEM corrected for within participant comparisons (*Cousineau, 2005*). Grey lines link thresholds per participant, per image type. For visualization, we have linked the bCFS and reCFS thresholds with broken lines, to better indicate that both do vary according to image category. (**b**) The relative difference in contrast between bCFS and reCFS thresholds is the same across image categories.

A 3x2 x 4 repeated measures ANOVA compared three rates of contrast change (slow, medium, fast), on both thresholds (bCFS, reCFS) across four image categories (face, object, grating, phase scrambled) using the tCFS paradigm. There was a significant main effect of threshold ($F(1,16) = 116.56$, p<0.001, $\eta_p^2 = 0.88$) again indicating that bCFS and reCFS contrasts differ. There was also a significant main effect of image type ($F(3,48) = 9.40$, p<0.001, $\eta_p^2 = 0.37$), again with no interaction with threshold (p=0.19). This result indicates that bCFS and reCFS thresholds vary in tandem regardless of image type. This null-effect of image-type was again confirmed with a Bayesian model comparison (3 speed x 4 image categories on suppression depth), demonstrating moderate support for the null effect of image category ($BF_{01} = 4.06$). Critically, there was a significant interaction between rate of contrast change and thresholds ($F(2,32) = 128.60$, p<0.001, $\eta_p^2 = 0.89$), as expected, indicating that the difference between bCFS and reCFS thresholds (i.e. suppression depth) depended on the target's rate of contrast change. *Figure 4* displays a summary of these results (averaged across image types), showing that as the rate of contrast change increases, so does suppression depth (Slow *M*=9.64 dB, SD = 4.37, Medium *M*=14.6 dB, SD = 5.43; Fast *M*=18.97 dB, SD = 6.93).

We performed additional analyses to rule out an alternative explanation for the increased suppression durations demonstrated in Experiment 3. We reasoned that if alternations between bCFS and reCFS were happening with a regular periodicity, such that responses were made every 1 second (for example), then these consistent responses could result in smaller suppression thresholds when the rate of contrast change was slow, as we have observed. Similarly, larger suppression thresholds would be measured if the same 1 s interval had elapsed while the rate of contrast change was fast. Inspection of the raw tCFS time-series qualitatively indicated that perceptual durations were varying with the rate of target contrast change (*Figure 4c*). To test this possibility, we compared the group average perceptual durations across all experiments, to test whether the average duration of percepts was the same despite different rates of contrast change. *Figure 4—figure supplement 1* displays the results of this analysis. The histograms of perceptual durations for Experiments 1 (all tCFS median M=3.83, SD = 1.66) and Experiment 2 (M=3.48, SD = 1.36) show similar means and distributions, with no significant difference between them (t(36) = 0.72, p=0.48). This is unsurprising given their similar design and matched rate of contrast change. In Experiment 3, however, the distribution of percept durations is shown to vary by rate of contrast change. With shorter median percept durations for fast rates of contrast change (M = 3.08, SD = 1.07), and slower percept durations for slow rates of contrast change (M=4.61, SD = 2.05) compared to medium (M = 3.45, SD = 1.29). A repeated measures ANOVA confirmed that median percept durations varied by rate of contrast change (F(2,32) = 30.89, p<0.001, $\eta p2=0.66$).

This analysis thus confirmed that these differences in suppression depth were not driven by fixed rates of perceptual alternation across the three levels of rate of contrast change (*Figure 4c* and *Figure 4—figure supplement 1*).

## Perceptual switches during tCFS are described by a damped harmonic oscillator

The results of Experiment 3 demonstrated that when the opportunity for target adaptation is increased, as when the target's rate of contrast change was slow, that suppression depth is reduced and a smaller contrast decrease is needed for a visible target to reenter CFS (see *Figure 4a*).One possible account for this relates to the balance of excitation/inhibition in neural systems, which have been particularly fruitful models of interocular competition (*Alais et al., 2010*; *Li et al., 2017*). In these models, adaptation over time is a critical parameter governing changes in visual consciousness (*Alais et al., 2010*), which motivated us to explore in our final analysis whether suppression depths also fluctuated over time. Accordingly, our final analysis sought to model the temporal nature of perceptual switches during tCFS, to understand whether a balance of excitation and inhibition may be contributing to the sequential contrast thresholds that govern target visibility.

For this analysis, a key dependent variable is the contrast difference between sequential thresholds (i.e. from bCFS to reCFS, or reCFS to bCFS). As the number of thresholds was the same in each trial, we averaged each threshold over observers to obtain a sequence of mean thresholds that preserved the order across the trial (e.g. average reCFS threshold at response 1, average bCFS threshold at response 2, average reCFS threshold at response 3 etc). Importantly, pooling across the threshold order rather than each participant's time series avoids smearing the data due to observers differing in

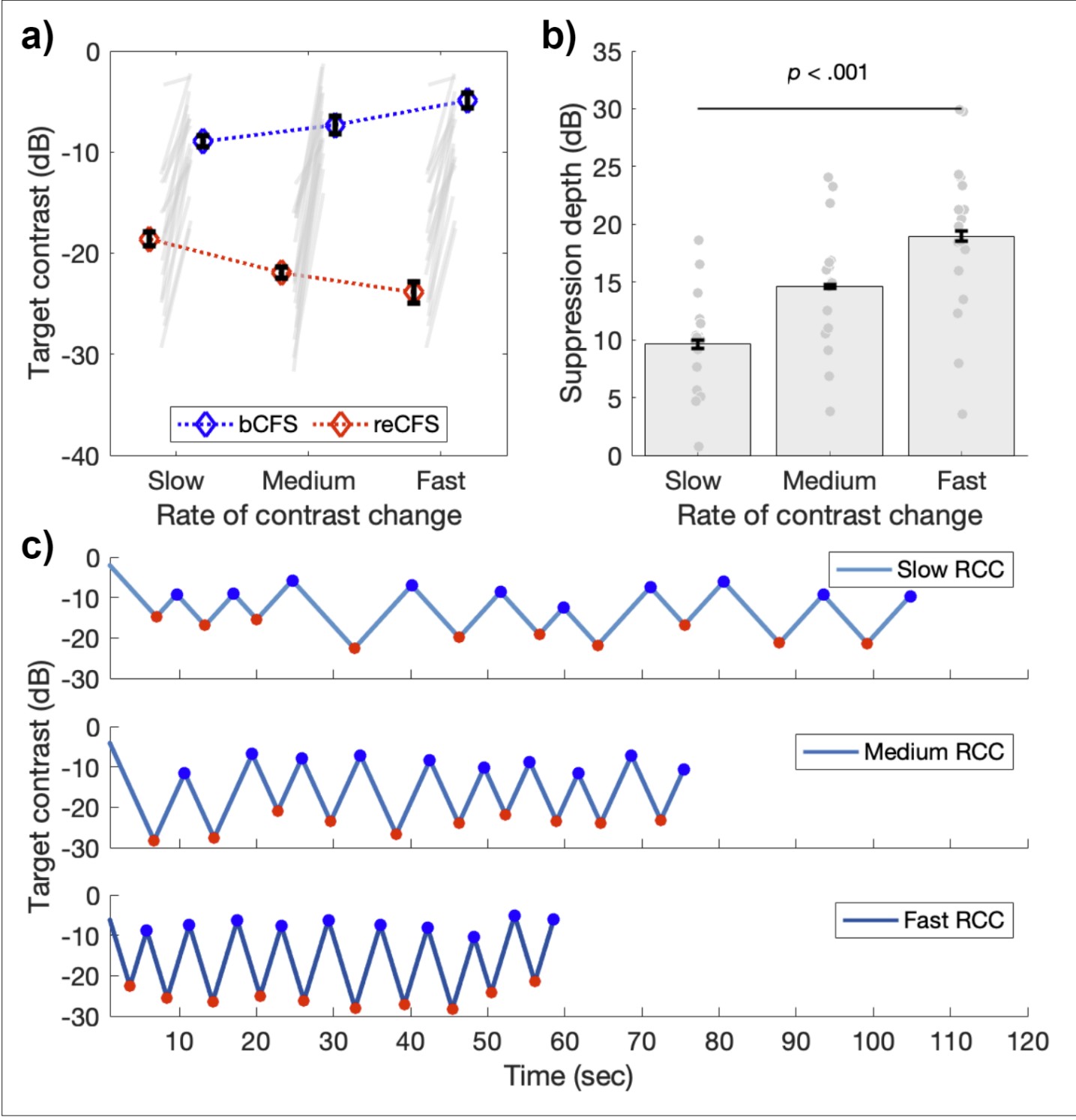

**Figure 4.** Suppression depth is greater with less time for adaptation. (**a**) bCFS (blue) and reCFS (red) thresholds collected during tCFS with three rates of contrast change (RCC) for the sample of 17 participants. Figure elements are the same as in *Figure 3a*. (**b**) Suppression depth increases when the rate of contrast change increases during tCFS. All error bars correspond to ± 1 SEM corrected for within participant comparisons (*Cousineau, 2005*). (**c**) Example trials at each rate of contrast change from a single participant. Red markers indicate reCFS responses, blue markers show bCFS.

The online version of this article includes the following figure supplement(s) for figure 4:

**Figure supplement 1.** Perceptual durations in Experiments 1–3.

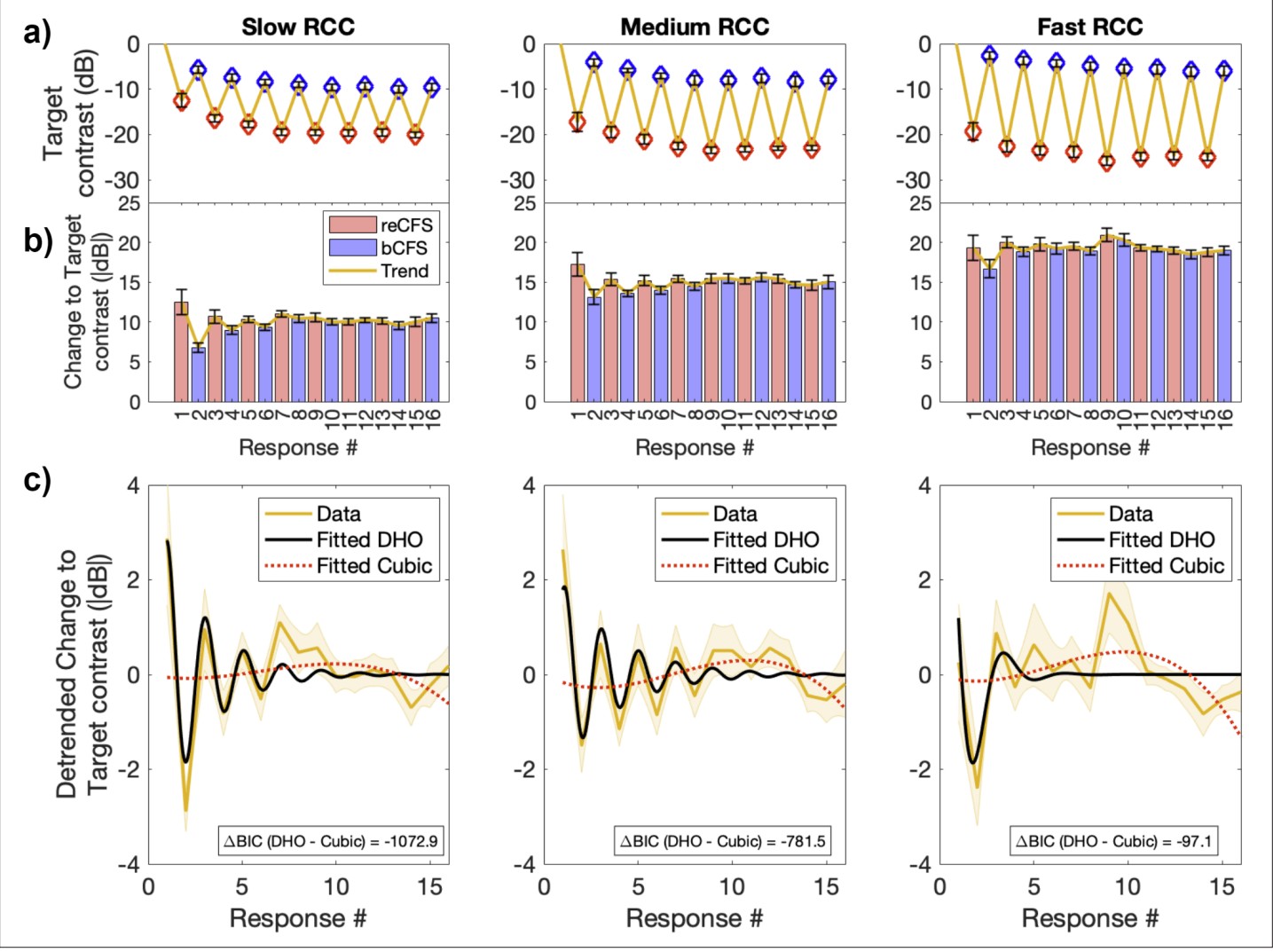

**Figure 5.** Suppression depth over time is well described by a damped harmonic oscillator. (**a**) Top row. Mean bCFS (blue) and reCFS (red) thresholds averaged by threshold order over the trial. Each column displays the average for slow, medium and fast rate of contrast change (RCC), respectively. (**b**) Middle row. Absolute change in target contrast between successive thresholds (i.e. suppression depth). Red bars indicate the decrease in target contrast needed for a visible target to reenter CFS. Blue bars indicate the increase in contrast for an invisible target to break CFS. (**c**) Bottom row. Yellow lines and shading plot the average change in suppression depth between successive thresholds. The data in each RCC condition is best fit by a damped harmonic oscillator (DHO) model shown in black. The change in Bayesian Information Criterion relative to the next best (cubic polynomial) model is displayed in each panel. All error bars and shading denote ± 1 SEM corrected for within participant comparisons (*Cousineau, 2005*).

their perceptual durations (*Figure 4—figure supplement 1*). *Figure 5a* displays the mean result separately for each target rate of change condition. To investigate the balance between bCFS and reCFS thresholds, we calculated the absolute change in contrast between sequential thresholds, as plotted in *Figure 5b*. These sequential estimates of suppression depth show marked fluctuations early in each trial, followed by a stabilisation to the grand average suppression depth for each rate of contrast change (Slow ~10 dB, Medium ~15 dB, Fast ~19 dB).

We next tested whether these sequential changes in suppression depth could be described by models of excitation and inhibition. For each rate of change condition, we compared the goodness of fit of three models: a simple harmonic oscillator, a damped harmonic oscillator, and a cubic polynomial model. We assessed the relative goodness of fit using the change in Bayesian Information Criterion (BIC) between models (see Materials and methods).

As can be appreciated in *Figure 5c*, sequential changes in suppression depth were well described by the damped harmonic oscillator model of excitation and inhibition, which well captured the

**Table 1.** Summary of results from model fits and comparisons.

| Model | Measure | Slow RCC | Medium RCC | Fast RCC |
|---|---|---|---|---|
| (cubic) 1 | $R^2$ | .07 | .19 | .29 |
| | BIC | −774.31 | 1343.79 | −944.23 |
| (harmonic) 2 | $R^2$ | .005 | .006 | 0 |
| | BIC | −706.85 | 1138.595 | −600.34 |
| (dHO) 3 | $R^2$ | .684 | .632 | .361 |
| | BIC | −1847.19 | −2125.29 | −1041.29 |
| Comparison | | | | |
| 2–1 | ΔBIC | 67.46 | 205.195 | 343.89 |
| 3–1 | ΔBIC | −1072.88 | −781.5 | −97.06 |
| 3–2 | ΔBIC | −1140.34 | −986.695 | −440.95 |

modulations early in the trial. For each rate of target contrast change, the damped harmonic oscillator model was the superior fit to the data (Slow contrast change, $R^2$=0.68, BIC = −1847.2; Medium contrast change, $R^2$=0.63, BIC = −2125.29; Fast contrast change, $R^2$=0.36, BIC = −1041.29). In each case, the change in Bayesian Information Criterion between the damped harmonic oscillator and the next best fitting cubic polynomial model was large (Slow contrast change, ΔBIC = −1072.88; Medium contrast change, ΔBIC = −781.5; Fast contrast change, ΔBIC = −97.06). *Table 1* displays the assessments of each model fit for each rate of change condition.

The damped harmonic oscillator has been proposed to capture the response of neural populations governed by excitation/inhibition balance after external perturbation, with a given rate of decay. We return to the interpretation of these models in the Discussion.

## Discussion

This study introduces a new CFS methodology that indexes suppression depth based on a participant's continuous report of the visibility status of a monocularly viewed target that steadily waxes and wanes in contrast, while the other eye views a CFS mask that forces the target into and out of suppression. The key to the technique, dubbed tracking CFS (tCFS), is that the participant's responses control when the reversals in the direction of contrast change occur: for example, a visible target initially decreases from high contrast until the participant reports the target's disappearance from awareness, at which time the target begins increasing in contrast until it again achieves awareness, and so on in an ongoing cycle. By tracking target visibility in this way, the method produces breakthrough thresholds (as in bCFS), but also yields measures of suppression thresholds(reCFS). By measuring the relative difference between these two thresholds, one can derive a rational index of suppression depth that, in turn, allows critical evaluation of whether suppression depth varies for different image categories (e.g. faces vs gratings). Experiment 1 introduced and validated the tCFS method for measuring suppression depth. Experiment 2 applied tCFS to images from different categories and found that suppression depth was invariant across a set of different types of images, regardless of their complexity, familiarity or salience. Experiment 3 manipulated suppression depth by varying the rate of target contrast change, providing presumptive evidence consistent with particular interpretation of the neural bases of the tCFS procedure. The following sections elaborate on each of these points.

The tCFS method offers two important advances for studying visual processing outside of awareness. First, it is fast and efficient. The data in *Figure 2b*, for example, show that 90 s is sufficient to collect a robust set of data comprising 16 thresholds, eight each for bCFS and reCFS. The second advantage is that by providing easy quantification of breakthrough and suppression thresholds, tCFS allows the calculation of a rational index of strength of suppression for evaluating whether certain categories of images (e.g. faces) receive prioritized processing, a controversial issue within the CFS literature. We will have more to say about claims of preserved visual processing despite suppression,

and claims of unconscious processing more generally in a moment (see *No effect of image category on suppression depth*).

## Thresholds for breaking and entering suppression quantify suppression depth

In Experiment 1, we used the tracking CFS method to measure bCFS and reCFS contrast thresholds, and compared them to those obtained using discrete unidirectional contrast changes – a series of measures with increasing target contrast to obtain bCFSthresholds as conventionally done, and also a series with decreasing target contrast to measure suppression (reCFS) thresholds. For both conditions, significant differences between bCFS and reCFS thresholds were observed. This difference indicates that suppression is not a passive process where a given contrast threshold determines a narrow awareness/suppression border. Instead, there is a contrast range (i.e. suppression depth) between awareness and suppression. This is consistent with models of interocular suppression involving mutually inhibitory left- and right-eye processes. The fact that binocular rivalry produces a significant degree of suppression depth is well known in that literature (*Blake and Camisa, 1979*; *Nguyen et al., 2003*) but is rarely considered in CFS studies (an exception to this rule is the study by *Tsuchiya et al., 2006* who used a briefly presented test probe to compare depth of suppression associated with CFS and with BR; suppression depth was considerably larger for the former than for the latter) where the focus is typically on the breakthrough threshold for awareness.

Intuitively, the average contrast threshold for reCFS should be lower than for bCFS, yet in the discrete conditions there were some participants who showed the inverse effect (n=4; *Figure 2c*). One possibility is that saccades, blinks, or other oculomotor reflexes that differentially impact transitions during interocular rivalry (*van Dam and van Ee, 2006a*; *van Dam and van Ee, 2006b*) might have influenced the discrete bCFS and reCFS trials at separate times. Another possibility is that by using a blocked design in the discrete trials, a participant's motivation or attention to task might have changed in the time elapsed between recording thresholds in bCFS and reCFS blocks. These possibilities highlight inherent advantages and controls of the tCFS procedure, for which no participant showed an inverse effect: bCFS and reCFS thresholds are recorded in alternation, thus each threshold judgment is relative to the previous judgment on the same image, recorded a matter of seconds (rather than minutes) prior. During tCFS, bCFS and reCFS tasks alternate, which helps maintain a level of engagement as the task is never repeated. Alternating between bCFS/reCFS tasks also means that any adaptation occurring over the trial will occur in close proximity to each threshold, as will any waning of attention. The benefit being that bCFS and reCFS thresholds are measured under closer conditions in continuous trials, compared to discrete ones. Long periods of target invisibility that can occur in typical bCFS studies when the target is raised from near-zero contrast are also avoided, as target contrast hovers around the visibility/invisibility thresholds. As *Figure 2c* shows, tCFS produces an increased size of suppression depth compared to discrete trials. Moreover, it can be administered quickly, and provides threshold measures with less variance.

As an aside, the existence of distinctly different, complementary transition thresholds for bCFS and reCFS is reminiscent of the behavior termed hysteresis: a property of dynamical systems wherein output values, rather than being solely governed by corresponding input values, also exhibit lags or delays based on the valence of continuous changes in the input values. Examples of hysteresis in visual perception include transitions between binocular fusion and binocular rivalry (*Anderson, 1992*; *Buckthought et al., 2008*; *Julesz and Tyler, 1976*), perception of motion direction in random-dot cinematograms (*Williams et al., 1986*), and repetition priming in perception of bistable configurations (*Pastukhov et al., 2015*). To paraphrase *Maglio and Polman, 2016*, hysteresis can be construed as a form of memory whereby prior states influence the persistence of current states into the future.

## No effect of image category on suppression depth

Having demonstrated a difference between bCFS and reCFS thresholds using the tCFS procedure, Experiment 2 compared suppression depth across five different image categories. As noted in the Introduction, a number of previous studies have interpreted differences in bCFS thresholds (or suppression durations) for different categories of images without taking into account thresholds for *inducing* suppression, which we believe is an important piece of information necessary for calculating suppression depth. In this paper, we applied the tCFS method to assess whether bCFS thresholds differed

among several image categories and, as well, whether category-specific differences in suppression depth would be obtained when reCFS thresholds were also included in the equation. As an important prerequisite for asking this question, our bCFS thresholds indeed did replicate the often reported finding that certain image types break out of suppression into awareness at lower contrasts than do others (see *Figure 3a*). For example, bCFS thresholds for faces were lower than bCFS thresholds for phase-scrambled images by 5.3 dB. Critically, however, while bCFS thresholds varied with image type, the reCFS thresholds for those and, indeed, for all images was approximately 15 dB lower than bCFS, *regardless of image type*. In other words, all five of the image categories we tested, regardless of complexity or familiarity, produced a constant depth of suppression averaging about 15 dB (see *Figure 3b*), even though their bCFS thresholds varied.

As a reminder, we explicitly tested image types that in other studies have shown differential susceptibility to CFS attributed to some form of expedited unconscious processing. Nevertheless, one could argue that our failure to obtain evidence for category specific suppression depth is based on the limited range of image categories sampled in this study. We agree it would be informative to broaden the range of image types tested using tCFS to include images varying in familiarity, congruence and affect. We can also foresee value in deploying tCFS to compare bCFS and reCFS thresholds for visual targets comprising physically meaningless 'tokens' whose global configurations can synthesise recognizable perceptual impressions. To give a few examples, dynamic configurations of small dots varying in location over time can create the compelling impression of rotational motion of a rigid, 3D object (structure from motion) or of a human engaged in given activity (biological motion) (*Grossmann and Dobbins, 2006*; *Watson et al., 2004*). These kinds of visual stimuli are associated with neural processing in higher-tier visual areas of the human brain, including the superior occipital lateral region (e.g. *Vanduffel et al., 2002*) and the posterior portion of the superior temporal sulcus (e.g. *Grossman et al., 2000*). These kinds of perceptually meaningful impressions of objects from rudimentary stimulus tokens are capable of engaging binocular rivalry. Such stimuli would be particularly useful in assessing high-level processing in CFS because they can be easily manipulated using phase-scrambling to remove the global percept without altering low-level stimulus properties. In a similar vein, small geometric shapes can be configured so as to resemble human or human-like faces, such as those used by *Zhou et al., 2021*. They derived dominance and suppression durations with fixed-contrast images. In their study, genuine face images and faux faces remained suppressed for equivalent durations whereas genuine faces remained dominant significantly longer than did faux faces. The technique used by those investigators - interocular flash suppression (*Wolfe, 1984*) - is quite different from CFS in that it involves abrupt, asynchronous presentation of dissimilar stimuli to the two eyes. It would be informative to repeat their experiment using the tCFS procedure. These kinds of faux faces could be used in concert with tCFS to compare suppression depth with that associated with actual faces.

Next, we turn to another question raised about our conclusion concerning invariant depth of suppression. If a certain image type had overall lower bCFS and reCFS contrast thresholds relative to another image type (despite equivalent suppression depth), would that imply the former image enjoyed 'preferential processing' relative to the latter? And, what would determine the differences in bCFS and reCFS thresholds? *Figure 3* shows that polar patterns tend to emerge from suppression at slightly lower contrasts than do gratings and that polar patterns, once dominant, tend to maintain dominance to lower contrasts than do gratings and this happens even though the rate of contrast change is identical for both types of stimuli. But while rate of contrast change is identical, the neural responses to those contrast changes may not be the same: neural responses to changing contrast will depend on the neural contrast response functions (CRFs) of the cells responding to each of those two types of stimuli, where the CRF defines the relationship between neural response and stimulus contrast. CRFs rise monotonically with contrast and typically exhibit a steeply rising initial response as stimulus contrast rises from low to moderate values, followed by a reduced growth rate for higher contrasts. CRFs can vary in how steeply they rise and at what contrast they achieve half-max response. CRFs for neurons in mid-level vision areas such as V4 and FFA (which respond well to polar stimuli and faces, respectively) are generally steeper and shifted toward lower contrasts than CRFs for neurons in primary visual cortex (which respond well to gratings). Therefore, the effective strength of the contrast changes in our tCFS procedure will depend on the shape and position of the underlying CRF, an idea we develop in more detail in Appendix 1, comparing the case of V1 and V4 CRFs. Interestingly, the

comparison of V1 and V4 CRFs shows two interesting points: (i) that V4 CRFs should produce much lower bCFS and reCFS thresholds than V1 CRFs, and (ii) that V4 CRFs should produce much more suppression than V1 CRFs. Our data do not support either prediction: bCFS and reCFS thresholds for the polar shape are not 'much lower' than those for gratings (*Figure 3*) and neither is there 'much more' suppression depth for the polar form. There is no room in these results to support the claim that certain images are special and receive 'preferential processing' or processing outside of awareness. Instead, the similar data patterns for all image types is most parsimoniously explained by a single mechanism processing all images (see Appendix 1), although there are many other kinds of images still to be tested in tCFS and exceptions may yet be found. As a first step in exploring this idea, one could use standard psychophysical techniques (e.g. *Ling and Carrasco, 2006*) to derive CRFs for different categories of patterns and then measure suppression depth associated with those patterns using tCFS.

In a similar vein, it's natural to wonder whether this non-selectivity of CFS depth of suppression applies to binocular rivalry suppression, too. *Blake and Fox, 1974* concluded that rivalry suppression is non-selective based on a task where observers were unable to notice large changes in the spatial frequency or orientation of a suppressed grating. On the other hand, *Alais and Melcher, 2007* found that the detectability of a brief, monocular probe presented to an eye during rivalry varied depending on the 'complexity' (e.g. grating vs face) of the stimulus being probed, with suppression being greater for complex images. *Tsuchiya et al., 2006* found that detection of a brief test probe was much more difficult to detect when presented to an eye during suppression phases of CFS compared to binocular rivalry. In a similar vein, durations of suppression phases associated with CFS are considerably longer than those associated with rivalry (~15 x longer, in the study by *Blake et al., 2019*). A clear next step will be to apply a variant of the tCFS paradigm to binocular rivalry to assess the uniformity of rivalry suppression depth based on stimulus complexity.

Turning to a methodological issue, an important caveat for interpreting suppression depth during tCFS is that bCFS thresholds will be slightly overestimated due to the brief reaction time delay between the perceived transition and execution of the motor response signalling that event (a problem for most bCFS studies). By the same token, reCFS thresholds will be slightly underestimated because of this response delay. These disparities between transition times and the responses to them lead to a slight inflation of suppression depth. For example, if we assume an average reaction time of 500ms for appearance and disappearance events, then suppression depth will be inflated by ~4.2 dB at the rate of contrast change used in Experiments 1 and 2 (0.07 dB per frame at 60 fps). This cannot account for suppression depth in its entirety, which was many times larger at approximately 14 dB across image categories. We have no reason to suspect that reaction-times would differ when reporting on the appearance or disappearance of different image categories under CFS, and indeed found no significant evidence for an interaction between thresholds and image categories in our analyses. This leaves a constant suppression depth of approximately 14 dB to be explained, and the value of comparing between image categories remains. In Experiment 3, the rate of contrast change varied which led to corresponding changes in suppression depth, which we note could also not be attributed to a reaction-time delay (*Figure 4—figure supplement 1*). Using the same assumptions of a 500ms response time delay, this would predict a suppression depth of 2.1 dB, 4.2 dB, and 6.3 dB for the slow, medium and fast ramp speeds respectively. However, this difference cannot account for the size of the effects (*Slow* 9.64 dB, *Medium* 14.6 dB, *Fast* 18.97 dB). The difference in suppression depth based on reaction-time delays (± 2.1 dB) also does not match with our empirical data (*Medium - Slow* = 4.96 dB; *Fast - Medium* = 4.37 dB).

Previous research has attributed faster CFS breakthrough (equivalently, lower contrast) to unconscious processing of suppressed images (*Gayet et al., 2014*; *Mudrik et al., 2011*). As the current study found uniform suppression depth for all tested images, even though bCFS thresholds varied, it is clear that differences in bCFS thresholds alone should not be interpreted in terms of expeditious unconscious processing of semantically relevant images. Indeed, if image categories such as faces were processed unconsciously, they reasonably should be harder to re-suppress, and thus have a smaller suppression depth compared to neutral stimuli (see *Figure 1*) – which was not the case.

As an alternative to lower bCFS thresholds being due to unconscious processing of images with relevant semantic content, such images (here, faces and objects) may break suppression at lower contrasts because they are particularly rich in low-level feature content that promotes image salience

(e.g. *Itti and Koch, 2000*). For example, faces and objects could be more salient due to peaks in local image contrast (*Parkhurst and Niebur, 2004*), contour integration (*Kapadia et al., 2000*), closed curvilinear form (concavity: *Schmidtmann et al., 2015*, and phase aligned spatial frequency spectra *Maehara et al., 2009*). The combination of these image properties plausibly render images such as faces and objects particularly salient within early stages of visual processing and thus lead to lower bCFS thresholds than those measured for less salient stimuli such as gratings or scrambled noise. Moreover, this derivation of salience is purely stimulus-based and arises without regard to semantic implications. This possibility, which has been raised by others (*Gayet et al., 2014*; *Moors and Hesselmann, 2018*; *Moors et al., 2016*; *Moors et al., 2017*; *Moors and Hesselmann, 2018*), has appeal on grounds of parsimony. With the advent of tCFS, it is now possible to measure bCFS and reCFS for salience-indexed stimuli to learn the extent to which stimulus-based salience impacts suppression depth.

Some caution is warranted here, however. It is not clear that all variation in bCFS thresholds can be explained by low-level image properties. There may be important high-level factors that also contribute to the salience of a given target image that make it visible at lower contrasts than other images (*Gayet et al., 2014*; *Jiang et al., 2007*; *Mudrik et al., 2011*; *Yang et al., 2007*). For example, faces provide essential social information and we are very highly attuned to them. Face images may therefore be salient at lower contrasts than other images such as random noise for high-level reasons specific to face processing. However, given there is a parsimonious account of low bCFS thresholds for faces in terms of low-level image properties, this would need to be established experimentally, perhaps using face inversion (*Matsuyoshi et al., 2015*) or contrast reversal (*Liu-Shuang et al., 2022*), manipulations that differentially impact visual processing of faces compared to other object categories This matter is still open and ripe for further experimental evidence, but our current findings add support to the argument that faces may become visible in CFS at lower contrasts for reasons unrelated to special access to awareness or partial processing during suppression.

A hybrid-model might therefore be needed for a full account of CFS, similar to those proposed for binocular rivalry (*Cao et al., 2021*; *Wilson, 2003*). Based on our results here, we imagine a hybrid model in which relative suppression depth for a given image arises from low-level interocular mutual inhibition acting equivalently on any kind of image (yielding the uniform suppression depth we observe), and the absolute level of breakthrough threshold could be impacted by high-level salience (e.g. *Hesse and Tsao, 2020*). Careful manipulation of low- and high-level properties, in combination with the tCFS method, should be able to test this model, as discussed earlier.

## Suppression depth is modulated by rate of contrast change

Experiment 3 varied the rate of target contrast change with the expectation that this would alter the magnitude of adaptation during tCFS. We predicted that a faster contrast change would reduce the opportunity for adaptation to accrue, thereby requiring a greater change in contrast to overcome suppression during CFS. Similarly, a slower rate of change should increase the opportunity for adaptation, resulting in the inverse effect. We observed strong modulations of suppression depth based on the rate of contrast change, confirming these predictions (*Figure 4*). Follow-up analyses confirmed that all three rates of contrast change had distinct percept duration times (*Figure 4—figure supplement 1*), indicating that the differences in suppression depth we observed were not due to an artefact such as participants responding with a fixed inter-response interval, which would spuriously increase suppression depth for a fast rate of change.

## Damped harmonic oscillator model

We modeled the changes in contrast between successive bCFS and reCFS thresholds over a trail and found that a damped harmonic oscillator (DHO) provided an excellent fit to these sequential estimates of suppression depth. The applicability of this model is noteworthy for a number of reasons. First, in neuroscience, the DHO model provides a valuable mathematical framework for understanding the dynamics of neural systems and their responses to external stimuli, particularly with regard to the interplay between excitation and inhibition (*Freeman, 1961*; *Hodgkin and Huxley, 1952*; *Spyropoulos et al., 2022*). In the present context, the high starting contrast of the suprathreshold target in tCFS trials is analogous to the external perturbation. The asymptotic differences in thresholds over time are reminiscent of both earlier (*Wilson, 2003*), and more recent computational models *Cao et al., 2021*

of interocular competition, models proposing that changes to visual awareness are driven by an out-of-equilibrium cortical network. We note that the locus of competing neural ensembles could reside in early visual stages (*Alais et al., 2010*; *Lankheet, 2006*; *Li et al., 2017*), late stages (*Hohwy et al., 2008*) or across hierarchical (*Cao et al., 2021*; *Wilson, 2003*) levels of visual processing. Although it is beyond the scope of the present work, future studies could vary the starting conditions of the tCFSprocedure, or manipulate higher-order influences such as attention and expectation to examine whether the return to equilibrium we have revealed conforms to the predictions of competing models.

### Future directions

The tCFS method equips researchers with a convenient method to measure bCFS and reCFS thresholds, and thus suppression depth. We have used tCFS here to establish that a uniform suppression depth exists across image categories, and that differences in bCFS thresholds alone cannot provide strong evidence for unconscious processing. Many substantive questions remain. For example, the depth of interocular suppression is reported to partially depend on spatial feature similarity between the competing images (*Alais and Melcher, 2007*; *Drewes et al., 2023*) and their temporal frequency (*Han et al., 2018b*; *Han and Alais, 2018a*). These factors could be parametrically varied to examine specifically whether they modulate bCFS thresholds alone, or whether they also cause a change in suppression depth by asymmetrically affecting reCFS thresholds. Previous findings can easily be revisited, such as results showing that bCFS varies with manipulations of semantic content (e.g. face inversion, or manipulating a face's emotion), results which form part of the claimed evidence for expedited unconscious processing of certain suppressed images.

### Conclusion

Across three experiments we have introduced the tCFS method and shown that traditional evidence for unconscious processing – based on differences in the threshold to reach awareness (bCFS threshold) – provide only half the story. Misleading conclusions about unconscious processing must be supported by measures of suppression depth, which can be calculated as the difference between both breakthrough (bCFS) and suppression (reCFS) thresholds. Using the tCFS method, we have measured these thresholds, and found uniform suppression depth across five image five categories. Notably, this uniform suppression depth is increased with reduced opportunity for target image adaptation, as is the case when target contrast changes rapidly. Collectively, the three tCFS experiments refute existing claims of high-level semantic information or target complexity influencing the depth of unconscious processing during interocular suppression. Future findings may yet confirm differences in suppression depth in certain circumstances, yet this will require measurement of both breakthrough and suppression thresholds to demonstrate the requisite changes in suppression depth.

## Materials and methods

### Participants

A total of 36 undergraduate psychology participants volunteered in exchange for course credit. All participated with informed consent and had normal or corrected-to-normal vision. Our sample size for Experiment 1 was based on power estimates to detect a moderate sized effect in a 2x2 repeated-measures design (*Faul et al., 2009*), while also exceeding the typical sample size used in bCFS studies to compensate for our novel paradigm (e.g. n=10, *Cha et al., 2019*, n=10–16, *Han et al., 2021*). We adjusted our power analysis after observing a strong effect size for the difference between bCFS and reCFS thresholds when using the tCFS method, resulting in fewer participants in Experiments 2 and 3. Experiment 1: N=20, (15 females), Experiment 2: N=18 (12 females), Experiment 3, N=17 (11 females). All participants in Experiment 3 also did Experiment 2, and one author (JC) also completed experiments 2 and 3. This study was approved by the University of Sydney Human Research Ethics Committee (HREC 2021/048).

### Apparatus

Visual stimuli were displayed on a Mac Pro (2013; 3.7 GHz Quad-Core Intel Xeon E5) computer, displayed on an Apple LED Cinema monitor (24 inch, 1920x1,200 pixel resolution, 60 Hz refresh rate), running OS X El Capitan (10.11.6). All experiments were programmed using custom MATLAB code,

and displayed using Matlab (ver R2017b) and Psychtoolbox (ver 3.0.13; *Brainard and Vision, 1997*). Responses were collected via the left mouse button of the right hand. A mirror stereoscope was used to partition participant's vision into separate left- and right-eye views, located approximately 51 cm from the screen, with a total optical path length of 57 cm.

## Stimuli

Participants dichoptically viewed a high-contrast Mondrian mask pattern (400x400 pixels, 7° x 7°) with one eye and a small target stimulus (130x130 pixels, 2.2° x 2.2°) with the other eye. Two binocularly presented white squares surrounded the mask and served as a fusion lock to maintain stable fusion, and each eye had a central fixation cross (18x18 pixels; 0.3° x 0.3°). The Mondrian pattern was grey-scale and consisted of overlapping circles of various sizes and intensities and was updated every fifth video frame (12 Hz). The mask's RMS contrast ranged between 0.07 and 0.09. As previous research has indicated that achromatic masks may be optimal to suppress achromatic targets (reviewed in *Pournaghdali and Schwartz, 2020*), we opted for grayscale Mondrian patterns to match our targets.

In all experiments, targets were viewed by the right eye and were selected at random from a set that was standardised in RMS contrast (20%) and mean luminance (before contrast ramping was applied). Target contrast was ramped up or down by scaling the target image's standard contrast within a range of.02–1.0. Importantly, all contrast scaling was done on a logarithmic scale in decibel units (i.e. conDb = 20 x log10(con)) to make the changing contrast effectively linear, thus coinciding with the visual system's logarithmic contrast response function. Minimum (.02) and maximum (1.0) contrast values were thus –33.98 and 0 dB, respectively, and contrast steps were.07 dB per video frame. In experiment 3, where the rate of target contrast change was manipulated, the contrast steps were.035,.07, or.105 dB units per video frame. Target location varied between trials, drawn from a uniform distribution of 200x200 pixels centred on the fixation cross.

## Procedure

Participants were given practice trials until they were familiar with CFS and the task used in these experiments. Participants were instructed to respond via mouse click the moment their subjective visibility of the target stimulus changed (either when a visible target became suppressed, or when a suppressed target became visible). Realizing that phenomenological quality of reversals may differ among participants (*Moors et al., 2017*; *Zadbood et al., 2011*), we encouraged participants in our study to establish a criteria for target appearance/disappearance in the practice session and to maintain it throughout the experiment. In all experiments, the dependent variable was the target contrast at the moment when a change in target visibility was judged to have occurred (either breaking suppression or succumbing to suppression).

## Experiment 1 - Using tCFS to measure suppression depth

Our first experiment tested two different procedures for measuring reCFS and bCFS thresholds: a continuous tracking procedure and a discrete trials procedure similar to that used in previous bCFS research. We hypothesised that a difference between bCFS and reCFS thresholds would provide evidence for a contrast range (i.e. suppression depth) between awareness and suppression, in contrast to the possibility of a given contrast threshold determining a narrow awareness/suppression border. By comparing the results between discrete and continuous methods, we sought to establish the feasibility of collecting multiple thresholds within a single-trial, allowing the rapid quantification of suppression depth in CFS paradigms. Experiment 1 compared CFS thresholds for targets (faces and objects) increasing in contrast and decreasing in contrast (i.e. bCFS and reCFS thresholds) in discrete trials and in continuous tracking trials, in a 2x2 repeated-measures, within-subjects design. In the discrete conditions, participants completed eight blocks of eight trials, during which target contrast always changed in one direction - either increasing from low to high as is typically done in bCFS studies to measure breakthrough thresholds, or decreasing from high contrast to low to measure a suppression threshold. In the continuous condition, the target contrast tracked down and up continuously, reversing direction after each participant response. Continuous trials always began with the target decreasing from maximum contrast so that the participant's first response was to report when it disappeared, which caused target contrast to increase until breakthrough was reported, which caused it to decrease again until suppression, etc.,. Continuous trials terminated after 16 reports of change in

target visibility. When the contrast time series is plotted as shown in *Figure 2b*, the plot shows eight upper turning points where the target broke into awareness (bCFS thresholds) and eight lower turning points where the target became re-suppressed (reCFS thresholds). The order of discrete and continuous blocks was counterbalanced and randomized across participants. Images for the eight trials of each block type were drawn from the same set of four faces and four naturalistic objects, with no repetitions within a block. Before each block began, participants completed a series of practice trials that utilised an independent set of six images. They were able to complete practice trials as many times as they wished until they had confidently established interocular fusion and were comfortable with the requirements of the task.

## Experiment 2 - The effect of image category

Experiment 2 tested the suppression depth obtained for different image categories. Experiment 2 used only the continuous tracking 'tCFS' method and compared five image types (faces, familiar objects, linear gratings, phase scrambled images, and polar patterns that were radial lines or concentric circles). The trials contained 20 reports (10 bCFS and 10 reCFS thresholds) and the data were analysed in a 5 (image type) x 2 (bCFS vs reCFS thresholds) within-subjects, repeated-measures ANOVA. There were 10 tracking trials, with each trial containing a single target image from a subset of ten (two of each image category, randomly ordered for each participant).

## Experiment 3 - Rate of contrast change on suppression depth

Experiments 1 and 2 introduced the tCFS method and demonstrated a uniformity of suppression depth across multiple target image categories. This uniformity of suppression depth could indicate that neural events mediating CFS suppression are not selective for complexity or semantic meaning, and like popular models of binocular rivalry, could instead be based on low-level reciprocal inhibition and neural adaptation processes (*Alais et al., 2010*; *Kang and Blake, 2010*; *Mcdougall, 1901*), in which case suppression depth should vary with the rate of contrast change of the monocular target. More specifically, based on the adapting mutual inhibition model, we predicted that at a slower rate of contrast change neural adaptation for the monocular target would increase, lowering the amount of contrast change necessary to transition between visibility states. Similarly, a faster rate of target contrast change would reduce the time for neural adaptation of the monocular target, resulting in an increase in the required change in contrast necessary to transition a target into and out of awareness. Expressed in operational terms, the depth of suppression should increase with the rate of target change, which was the focus of Experiment 3.

Experiment 3 used the tCFS method and compared the rate of target contrast change (slow, medium, fast) across four image categories (faces, objects, linear gratings, and phase scrambled images). The trials contained 20 reports (10 bCFS and 10 reCFS thresholds) and the data were analysed in a 3 (contrast change rate) x 4 (image type) x 2 (bCFS vs reCFS thresholds) repeated-measures, within-subjects design. The medium rate of change was the same as used in Experiments 1 and 2, and the slow and fast rates were 0.5 and 1.5 times the medium rate, respectively. There were 12 tracking trials, given by the factorial combination of four target image types repeated at the three rates of target contrast change (in a randomised order for each participant).

## Data analysis

Data analysis was performed in Matlab (ver R2022a), and SPSS/JASP (ver 28). Initial inspection identified 1 participant for exclusion (from Experiment 3), based on failure to follow task instructions. For visualization and analysis, all contrast thresholds are expressed in decibel units.

We performed Bayesian model comparison to quantify evidence for and against the null in JASP, using Bayesian repeated measures ANOVAs (uninformed prior with equal weight to all models). We report Bayes factors (*B*) for main effects of interest (e.g. effect of image type on suppression depth), as evidence in favor compared to the null model ($BF_{10}=B$). Following the guidelines recommended in *Dienes, 2021*, *B* values greater than 3 indicate moderate evidence for H1 over H0, and *B* valuesless than1/3 indicate moderate evidence in favor of the null. *B* values residing between 1/3 and 3 are interpreted as weak evidence, or an insensitivity of the data to distinguish between the null and alternative models.

## Model fitting

We additionally quantified the change in relative contrast over time, and evaluated a series of model fits to describe these data. For this analysis, bCFS and reCFS thresholds were first averaged within their respective response number, enabling a comparison of thresholds over the course of each trial. The modelling used the absolute change in contrast between each sequential threshold in the tracking series as its dependent variable, which we modeled after detrending, per participant, and at the group level.

We compared three basic models to this data. All models were fit with a non-linear least-squares approximation using a maximum of 400 iterations (lsqcurvefit.m in MATLAB). The first was a simple cubic polynomial with three free parameters:

$$ax^3 + bx^2 + cx \tag{1}$$

where a, b and c are coefficients for the cubic, quadratic, and linear term. We also fit a simple harmonic oscillator with three free parameters:

$$a \times \sin(b \times t + c) \tag{2}$$

where a is amplitude, b is frequency, c is a phase offset, and t is time. We also fit a damped harmonic oscillator model with four free parameters:

$$a \times e^{-b \times t} \times \sin(c \times t + d) \tag{3}$$

where a and b describe the amplitude and damping coefficient of decay, and c and d describe the frequency and phase shift of the oscillatory response.

To fit each model, we linearly interpolated between the turning points (thresholds) in the tCFS time series of each trial to increase the observations to 1000 samples, and estimated the goodness of each fit through a series of steps. First, we calculated the sum of squared residual errors for each fit (SSE), and calculated the Bayesian Information Criterion (BIC) using *Equation 4*:

$$BIC = n \times \log(SSE/n) + k \times \log(n); \tag{4}$$

where n represents the number of observations in the dataset, SSE is the sum of squared errors, and k is the number of parameters in the model. The BIC allows a comparison of model fits while taking into account the goodness of fit and complexity of each model. It includes a penalty on the number of parameters in the model by including a term that scales with the logarithm of sample size. When comparing two models, the model with a lower BIC is considered favorable, with a change of 0–2 BIC as weak evidence in favor, and 6–10 as strong evidence in favor (*Kass and Raftery, 1995*).

## Acknowledgements

RB was supported by funds from a Centennial Professorship Research Award, Vanderbilt University, with thanks to Jeffrey Schall. DA and MD were supported by an Australian Research Council grant (DP210101691).

## Additional information

### Funding

| Funder | Grant reference number | Author |
| --- | --- | --- |
| Australian Research Council | DP210101691 | David Alais |

The funders had no role in study design, data collection and interpretation, or the decision to submit the work for publication.

## Author contributions
David Alais, Conceptualization, Resources, Software, Supervision, Investigation, Writing – original draft, Project administration, Writing – review and editing; Jacob Coorey, Data curation, Investigation, Methodology, Writing – original draft, Project administration, Writing – review and editing; Randolph Blake, Writing – original draft, Writing – review and editing; Matthew J Davidson, Conceptualization, Formal analysis, Supervision, Validation, Investigation, Visualization, Methodology, Writing – original draft, Writing – review and editing

## Author ORCIDs
David Alais (ID) https://orcid.org/0000-0002-0411-940X
Randolph Blake (ID) https://orcid.org/0000-0001-8697-6239
Matthew J Davidson (ID) https://orcid.org/0000-0002-2088-040X

## Ethics
This study was approved by the University of Sydney Human Research Ethics Committee (HREC 2021/048). All participants provided informed consent prior to participation.

Reviewer #1 (Public review): https://doi.org/10.7554/eLife.91019.4.sa1
Reviewer #2 (Public review): https://doi.org/10.7554/eLife.91019.4.sa2
Reviewer #3 (Public review): https://doi.org/10.7554/eLife.91019.4.sa3
Author response https://doi.org/10.7554/eLife.91019.4.sa4

---

# Additional files

## Supplementary files
• MDAR checklist

## Data availability
All data and analysis codes are available in a public repository on the open science framework: https://osf.io/nzp9v/. Analysis code to reproduce the main results and figures are available in a public repository https://github.com/Davidson-MJ/tCFS-elife-VOR (copy archived at *Davidson, 2024*).

The following dataset was generated:

| Author(s) | Year | Dataset title | Dataset URL | Database and Identifier |
| --- | --- | --- | --- | --- |
| Davidson M | 2023 | tCFS: tracking paradigm (Target contrast ramps, v1) | https://doi.org/10.17605/OSF.IO/NZP9V | Open Science Framework, 10.17605/OSF.IO/NZP9V |

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

## Appendix 1

### Difference thresholds as a measure of suppression depth and the role of contrast response functions

One question raised in review concerned using a decibel contrast scale and the calculation of suppression depth as the difference between breakthrough and suppression thresholds. The justification for this is twofold. First, neurons in early visual cortex exhibit a logarithmic-like response to contrast that grows rapidly at low contrasts and then saturates at higher contrasts. Data from single V1 neurons show that these contrast response functions (CRFs) to linear contrast can be very adequately described by fitting a logarithmic curve, but not by linear or power functions (*Albrecht and Hamilton, 1982*). Consistent with a logarithmic response, perceptual thresholds for contrast increments tend to be a constant proportion of the base contrast (i.e. Weber's law holds). For these reasons, logarithmically scaled contrast changes should therefore produce approximately linear changes in perceived contrast and should be used when ramping contrast in CFS procedures. Second, once contrast is log-transformed to decibels, the difference calculation used to obtain suppression depth is effectively a ratio. It is edifying to learn, as we show here, that suppression depth is constant as this indicates that whatever value the bCFS threshold has, the reCFS threshold will be *n* times lower. For example, a difference of –12 dB is a factor of four (e.g. bCFS = 0.2, reCFS = 0.05). It is important, though, to plot bCFS and reCFS values as well as suppression depth because these thresholds could be higher or lower for a different image category and yet still produce the same suppression depth (e.g. values of bCFS = 0.6 and reCFS = 0.15 also produce a suppression depth of –12 dB but are very different in absolute terms). Plotting bCFS and reCFS thresholds and suppression depth ensures absolute shifts are not obscured.

It is interesting to consider that the shapes of CRFs vary at different stages of cortical processing. CRFs tend to be shallower in V1 than in subsequent areas such as V2, V4, and FFA where they are considerably steeper. *Appendix 1—figure 1* plots two CRFs, modeled with Naka-Rushton functions, showing a steep and a shallow version. The Naka-Rushton equation provides a very good description of the CRF (*Albrecht and Hamilton, 1982*) and is the preferred way to model them. The shallower CRF typifies that measured from neurons in primary visual cortex. It has a C50 (contrast giving half-max response) of 0.24 and an exponent of 3.4 – the mean values of a sample of 98 primate V1 CRFs reported by *Albrecht and Hamilton, 1982*. The steeper CRF is typical of a V4 CRF and has a C50 of 0.15 and an exponent of 1.9 – the mean values of 131 V4 CRFs reported by *Cheng et al., 1994*; *Williford and Maunsell, 2006*. For the sake of illustrating a key point, we have added horizontal lines labeled bCFS and reCFS threshold. We make the simple assumption that when contrast is ramping up, a target will break through from suppression to visibility when response level reaches the 'bCFS threshold'. Conversely, a visible target ramping down in contrast will become suppressed at the 'reCFS threshold'.

The vertical lines in *Appendix 1—figure 1* show the bCFS/reCFS threshold on the contrast axis, revealing a smaller range of 0.24–0.33 for the V1 CRF than for the V4 CRF (0.15–0.265). When expressed in decibels (SDdB = 20 * log10(reCFS/bCFS)) these thresholds produce suppression depths of 2.77 dB for the V1 CRF and 4.94 dB for the V4 CRF. This implies that CFS suppression should be deeper if it were occurring at the level of V4, and shallower if it were occurring in V1.

The suppression values calculated from the figure are smaller than those we report here in our data. However, it is important to note that the example CRFs in *Appendix 1—figure 1* are taken from recordings of single neurons, whereas an individual percept, such as a grating emerging into visibility, will be the result of a whole population of neurons. This is significant because when CRFs are derived from pooled neural responses using brain imaging techniques such as EEG or MEG population CRFs are much broader (*Avidan et al., 2002*; *Hall et al., 2005*). Importantly they broaden yet still maintain the same pattern as plotted in *Appendix 1—figure 1* where V1 CRFs show a more graded response to contrast and CRFs for areas beyond V1 show left-shifted and steeper CRFs. Therefore, the broader CRFs at the population level would produce much bigger bCFS/reCFS threshold differences and consequently a greater suppression depth that would be closer to the values we report here.

Outlining a realistic model of CRFs not only adds clarity about one of the key mechanisms underlying the contrast ramping procedure used in tCFS, it also allows us to make predictions and test hypotheses. For example: does CFS suppression depend on the type of target image? Neurons in area V4 are selective for polar form (*Gallant et al., 1996*) but V1 neurons are not. V1 neurons, on

the other hand, are highly selective for linear gratings. Neither area is selective for faces, yet area FFA is specialised for this (*Bao and Tsao, 2018*) and shares similar CRFs to V4. If CFS suppression were to occur in the area specialised for a given target stimulus, our CRF model makes two predictions: (i) that the bCFS and reCFS thresholds should be very different when gratings are compared to polar gratings or faces (V1 bCFS/reCFS thresholds should be higher), and (ii) suppression depth should be greater for polar gratings and faces than for gratings. However, neither of these predictions is supported by our data. *Figure 3a* shows clearly that across different kinds of target images bCFS and reCFS thresholds tend to be very similar. Indeed, post hoc t-tests comparing gratings with objects, faces, and polar gratings show no significant differences between bCFS thresholds (all ps >0.572) or reCFS thresholds (all ps >0.285). Moreover, the level of suppression depth is the same (*Figure 3b*). This pattern is not consistent with target images being suppressed in their preferred cortical areas but with suppression occurring in a single area governed by a common CRF. Based on our findings, we suggest the critical single area for CFS suppression is most likely area V1 because: (i) this is the first cortical area where interocular mismatches are detected, and (ii) regardless of image type, all target images tested in our study are very much mismatched with the Mondrian mask and would likely trigger interocular suppression at this first binocular stage. It remains a possibility that future work will uncover evidence of higher level areas feeding into CFS suppression in a kind of hybrid model that has found support in studies of binocular rivalry suppression (*van Boxtel et al., 2008*; *Wilson, 2003*). One way to do that might be to create targets and masks that are congruent in as many features as possible to circumvent a front-end suppression triggered by an initial interocular mismatch.

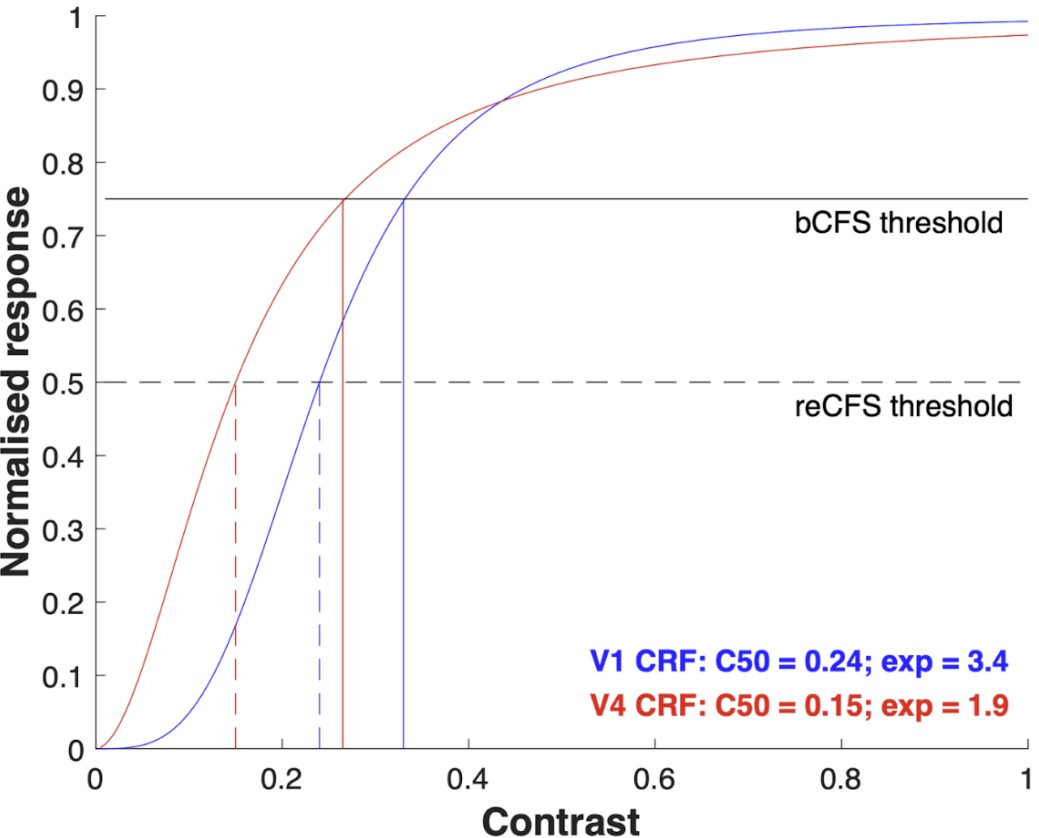

**Appendix 1—figure 1.** Contrast response functions and CFS thresholds. Two Naka-Rushton functions modelling contrast response functions (CRFs) are shown, one has parameters typical of primary visual cortical neurons (blue) and the other is typical of CRFs found in area V4 (red). The C50 and exponent values are taken from single-unit neurophysiological studies of primate areas V1 and V4. We make the simple assumption that when contrast is ramping up, a target will break through from suppression to visibility when response level reaches the 'bCFS threshold'. Conversely, a visible target ramping down in contrast will become suppressed at the 'reCFS threshold'. *Appendix 1—figure 1 continued on next page*

*Appendix 1—figure 1 continued*

Note that the V4 CRF has bCFS/reCFS thresholds that are much lower than those from the V1 CRF, and that the bCFS/reCFS range is greater for the V4 CRF, implying greater suppression depth (see appendix text). Our finding of constant suppression depth and very similar bCFS/reCFS thresholds across diverse image types suggests that CFS suppression for all our tested target images occurs in a common mechanism likely to be early in binocular processing.

