## [Editor Report · eLife assessment]

This **valuable** study introduces an innovative method for measuring interocular suppression depth, which implicates mechanisms underlying subconscious visual processing. The evidence is **solid** in suggesting that the new method yields provocative uniform suppression depth results across image categories that differ from conventional bCFS threshold. It will be of interest not only to cognitive psychologists and neuroscientists who study sensation and perception but also to philosophers who work on theories of consciousness.

---

## [Referee Report · Reviewer #1 (Public review)]

Summary

A new method, tCFS, is introduced to offer richer and more efficient measurement of interocular suppression. It generates a new index, the suppression depth, based on the contrast difference between the up-ramped contrast for the target to breakthrough suppression and the down-ramped contrast for the target to disappear into suppression. A uniform suppression depth regardless of image types (e.g., faces, gratings and scrambles) was discovered in the paper, favoring an early-stage mechanism involving CFS. Discussions about claims of unconscious processing and the related mechanisms.

Strength

The tCFS method adds to the existing bCFS paradigms by providing the (re-)suppression threshold and thereafter the depression depth. Benefiting from adaptive procedures with continuous trials, the tCFS is able to give fast and efficient measurements. It also provides a new opportunity to test theories and models about how information is processed outside visual awareness.

Weakness:

This paper reports the surprising finding of uniform suppression depth over a variety of stimuli. This is novel and interesting. But given the limited samples being tested, the claim of uniformity suppression depth needs to be further examined, with respect to different complexities and semantic meanings.

From an intuitive aspect, the results challenged previous views about ‘preferential processing’ for certain categories, though it invites further research to explore what exactly could suppression depth tell us about unconscious visual processing.

---

## [Referee Report · Reviewer #2 (Public review)]

Summary

The paper concerns the phenomenon of continuous flash suppression (CFS), relevant to questions about the extent and nature of subconscious visual processing. Whereas standard CFS studies only measure the breakthrough threshold-the contrast at which an initially suppressed target stimulus with steadily increasing contrast becomes visible-this study also measures the re-suppression threshold, the contrast at which a visible target with decreasing contrast becomes suppressed. Thus, the authors could calculate suppression depth, the ratio between the breakthrough and re-suppression thresholds. To measure both thresholds, the study introduces the tracking-CFS method, a continuous-trial design that results in faster, better controlled, and lower-variance threshold estimates compared to the discrete trials standard in the literature. The study finds that suppression depths are similar for different image categories, providing an interesting contrast to previous results that breakthrough thresholds differ for different image categories. The new finding calls for a reassessment of interpretations based solely on the breakthrough threshold that subconscious visual processing is category-specific.

Strengths

(1) The tCFS method quickly estimates breakthrough and re-suppression thresholds using continuous trials, which also better control for slowly varying factors such as adaptation and attention. Indeed, tCFS produces estimates with lower across-subject variance than the standard discrete-trial method (Fig. 2). The tCFS method is straightforward to adopt in future research on CFS and binocular rivalry.

(2) The CFS literature has lacked re-suppression threshold measurements. By measuring both breakthrough and re-suppression thresholds, this work calculated suppression depth (i.e., the difference between the two thresholds), which warrants different interpretations from the breakthrough threshold alone.

(3) The work found that different image categories show similar suppression depths, suggesting some aspects of CFS are not category-specific. This result enriches previous findings that breakthrough thresholds vary with image categories. Re-suppression thresholds vary symmetrically, such that their differences are constant.

Weakness

I do not follow the authors' reasoning as to why the suppression depth is a better (or fuller, superior, more informative) indication of subconscious visual processing than the breakthrough threshold alone. To my previous round of comments, the authors replied that 'breakthrough provides only half of the needed information.' I do not understand this. One cannot infer the suppression depth from the breakthrough threshold alone, but *one cannot obtain the breakthrough threshold from the suppression depth alone*, either. The two measures are complementary. (To be sure, given *both* the suppression depth and the re-suppression threshold, one can recover the breakthrough threshold. The discussion concerns the suppression depth *alone* and the breakthrough threshold *alone*.) I am fully open to being convinced that there is a good reason why the suppression depth may be more informative than the breakthrough threshold about a specific topic, e.g., inter-ocular suppression or subconscious visual processing. I only request that the authors make such an argument explicit. For example, in the significance statement, the authors write, 'all images show equal suppression when both thresholds are measured. We *thus* find no evidence of differential unconscious processing and *conclude* reliance on breakthrough thresholds is misleading' (emphasis added). Just what supports the 'thus' and the 'conclude'? Similarly, at the end of the introduction, the authors write, '[...] suppression depth was constant for faces, objects, gratings and visual noise. *In other words*, we find no evidence to support differential unconscious processing among these particular, diverse categories of suppressed images' (emphasis added). I am not sure the statements in the two sentences are equivalent.

The authors' reply included a discussion of neural CRFs, which may explain why the bCFS thresholds differ across image categories. A further step seems necessary to explain why CRFs do not qualify as a form of subconscious processing.

---

## [Referee Report · Reviewer #3 (Public review)]

Summary:

In the 'bCFS' paradigm, a monocular target gradually increases in contrast until it breaks interocular suppression by a rich monocular suppressor in the other eye. The present authors extend the bCFS paradigm by allowing the target to reduce back down in contrast until it becomes suppressed again. The main variable of interest is the contrast difference between breaking suppression and (re) entering suppression. The authors find this difference to be constant across a range of target types, even ones that differ substantially in the contrast at which they break interocular suppression (the variable conventionally measured in bCFS). They also measure how the difference changes as a function of other manipulations. Interpretation is in terms of the processing of unconscious visual content, as well as in terms of the mechanism of interocular suppression.

Strengths:

Interpretation of bCFS findings is mired in controversy, and this is an ingenuous effort to move beyond the paradigm's exclusive focus on breaking suppression. The notion of using the contrast difference between breaking and entering suppression as an index of suppression depth is interesting. The finding that this difference is similar for a range of target types that do differ in the contrast at which they break suppression, suggests a common mechanism of suppression across those target types.

---

## [Author Response]

The following is the authors’ response to the previous reviews.

We thank the editorial team and reviewers for their continued contributions to improve our work.

Below we have addressed the final recommendations to the authors

**Recommendations for the authors:**

**Reviewer #2 (Recommendations For The Authors):**
I asked previously why the suppression depth should vary based on the contrast change speed. I now understand that the authors expect this variation from a working model based on neural adaptation (lines 274-277 and 809-820). I suggest the authors specify this prediction also on lines 473-479, where there is room for improved clarity (the words/phrases 'impact,' 'be sensitive to,' and 'covary' are non-directional).

We have now specified this prediction to improve clarity:

Line 475 – 486

‘In the context of the tCFS method, the steady increases and decreases in the target’s actual strength (i.e., its contrast) should, respectively, boost its emergence from suppression (bCFS) and facilitate its reversion to suppression (reCFS) as it competes against the mask. Whether construed as a consequence of neural adaptation or error signal, we surmise that these cycling state transitions defining suppression depth should be sensitive to the rate of contrast change of the monocular target. Specifically, the slower the contrast change, the greater the amount of accrued adaptation, which will contract the range between breakthrough and suppression thresholds according to an adapting reciprocal inhibition model. For fast contrast change, there will be less accrual of adaptation meaning that the range between breakthrough and suppression thresholds will exhibit less contraction. Expressed in operational terms, the depth of suppression should be positively related to the rate of target change. Experiment 3 tested this supposition using three rates of contrast change.’

Line 108: 'By comparing the thresholds for a target to transition into (reCFS) and out of awareness (bCFS)'-are 'into' and 'out of' reversed?

They were, thank you, these have now been corrected.

Lines 696-698 read, 'Figure 3 shows that polar patterns tend to emerge from suppression at slightly lower contrasts than do gratings.' In the same paragraph, lines 716-171 read, 'Figure 3 shows that bCFS and reCFS thresholds are very similar for all image categories.' There is a statistically significant effect of category in these results; meanwhile, the differences among categories are arguably small. Which side do the authors intend to emphasize? Are the readers meant to interpret this as a glass-half-full, half-empty situation?

We have now revised this paragraph. We emphasise that the small differences do not support ‘preferential processing’ of the magnitude that would be expected from category specific neural CRFs.

From Line 702

‘Next we turn to another question raised about our conclusion concerning invariant depth of suppression. If a certain image type had overall lower bCFS and reCFS contrast thresholds relative to another image type (despite equivalent suppression depth), would that imply the former image enjoyed ‘preferential processing’ relative to the latter? And, what would determine the differences in bCFS and reCFS thresholds? Figure 3 shows that polar patterns tend to emerge from suppression at slightly lower contrasts than do gratings and that polar patterns, once dominant, tend to maintain dominance to lower contrasts than do gratings and this happens even though the rate of contrast change is identical for both types of stimuli. But while rate of contrast change is identical, the neural responses to those contrast changes may not be the same: neural responses to changing contrast will depend on the neural contrast response functions (CRFs) of the cells responding to each of those two types of stimuli, where the CRF defines the relationship between neural response and stimulus contrast. CRFs rise monotonically with contrast and typically exhibit a steeply rising initial response as stimulus contrast rises from low to moderate values, followed by a reduced growth rate for higher contrasts. CRFs can vary in how steeply they rise and at what contrast they achieve half-max response. CRFs for neurons in mid-level vision areas such as V4 and FFA (which respond well to polar stimuli and faces, respectively) are generally steeper and shifted towards lower contrasts than CRFs for neurons in primary visual cortex (which respond well to gratings). Therefore, the effective strength of the contrast changes in our tCFS procedure will depend on the shape and position of the underlying CRF, an idea we develop in more detail in Supplementary Appendix 1, comparing the case of V1 and V4 CRFs. Interestingly, the comparison of V1 and V4 CRFs shows two interesting points: (i) that V4 CRFs should produce much lower bCFS and reCFS thresholds than V1 CRFs, and (ii) that V4 CRFs should produce much more suppression than V1 CRFs. Our data do not support either prediction: bCFS and reCFS thresholds for the polar shape are not ‘much lower’ than those for gratings (Fig. 3) and neither is there ‘much more’ suppression depth for the polar form. There is no room in these results to support the claim that certain images are special and receive ‘preferential processing’ or processing outside of awareness. Instead, the similar data patterns for all image types is most parsimoniously explained by a single mechanism processing all images (see Appendix 1), although there are many other kinds of images still to be tested in tCFS and exceptions may yet be found. As a first step in exploring this idea, one could use standard psychophysical techniques (e.g., (Ling & Carrasco, 2006)) to derive CRFs for different categories of patterns and then measure suppression depth associated with those patterns using tCFS.’